# COVID-19: Has the Liver Been Spared?

**DOI:** 10.3390/ijms24021091

**Published:** 2023-01-06

**Authors:** Nicolò Brandi, Daniele Spinelli, Alessandro Granito, Francesco Tovoli, Fabio Piscaglia, Rita Golfieri, Matteo Renzulli

**Affiliations:** 1Department of Radiology, IRCCS Azienda Ospedaliero-Universitaria di Bologna, Via Albertoni 15, 40138 Bologna, Italy; 2Division of Internal Medicine, Hepatobiliary and Immunoallergic Diseases, IRCCS Azienda Ospedaliero-Universitaria di Bologna, 40138 Bologna, Italy

**Keywords:** COVID-19, liver, hepatocellular carcinoma, chronic liver disease, management

## Abstract

The liver is a secondary and often collateral target of COVID-19 disease but can lead to important consequences. COVID-19 might directly cause a high number of complications in patients with pre-existing chronic liver disease, increasing their risk of hepatic decompensation. Moreover, it also determines indirect consequences in the management of patients with liver disease, especially in those suffering from decompensated cirrhosis and HCC, as well as in the execution of their follow-up and the availability of all therapeutic possibilities. Liver imaging in COVID-19 patients proved to be highly nonspecific, but it can still be useful for identifying the complications that derive from the infection. Moreover, the recent implementation of telemedicine constitutes a possible solution to both the physical distancing and the re-organizational difficulties arising from the pandemic. The present review aims to encompass the currently hypothesized pathophysiological mechanisms of liver injury in patients with COVID-19 mediated by both the direct invasion of the virus and its indirect effects and analyze the consequence of the pandemic in patients with chronic liver disease and liver tumors, with particular regard to the management strategies that have been implemented to face this worldwide emergency and that can be further improved.

## 1. Introduction

Although it has been more than 2 years since the first outbreak, the coronavirus disease 2019 (COVID-19) pandemic is still having a profound and devastating impact on global healthcare systems. COVID-19 has a wide range of clinical presentations, varying from asymptomatic or mildly symptomatic infection to severe bilateral pneumonia, with a high risk of developing acute respiratory distress syndrome (ARDS) (20–67%) and thus the need for mechanical ventilation and Intensive Care Unit (ICU) admission.

Despite viral pneumonia representing the most common serious manifestation of COVID-19, extrapulmonary manifestations of COVID-19 have progressively gained attention due to their links to clinical outcomes and their potential long-term sequelae, especially in critically ill patients [1]. Vascular complications, myocardial dysfunction, acute kidney injury, gastrointestinal symptoms, neurologic complications, and dermatologic conditions are among the reported extrapulmonary complications [2,3]. Furthermore, recent studies have suggested that COVID-19 could also have a serious impact on the reproductive system, altering male sperm parameters and increasing the rate of female gestational disorders, such as preeclampsia [4,5]. However, whether COVID-19 could also directly affect the liver has been debated, and the literature regarding hepatic involvement in COVID-19 patients is heterogenous, due to variability in the definitions of liver dysfunction and differences in the clinical presentation and disease severity [6].

According to the current data, liver dysfunction or injury, defined as liver test abnormalities, has been reported with a prevalence of approximately 25% in COVID-19 patients, ranging from approximately 3% to 60% [6,7,8]. In particular, several studies demonstrated that patients with COVID-19 who develop liver dysfunction are mostly male, elderly, and obese [9,10,11,12]. Furthermore, hepatic dysfunction is significantly higher in critically ill patients, reaching up to 45% of cases, and is associated with a poor outcome, underlining its importance in clinical settings [13,14]. In those patients with already impaired liver function, SARS-CoV-2 infection may be responsible for the worsening of underlying chronic liver disease [15]. On top of that, COVID-19 may also exert an indirect effect on these patients, disrupting their care as a result of the failure of the screening, treatment, and follow-up [16].

Despite all the evidence, the pathophysiological and immunological mechanisms of liver injury in patients with COVID-19 are still poorly understood, as well as their long-term sequelae. Moreover, the consequences of its indirect effects on the management of patients with chronic liver disease are still emerging. Therefore, with the now inevitable certainty of having to live with this virus and its new variants at least for the next few years, clinicians must pay high attention to the most exposed and fragile patients and continuously search for new strategies that can be implemented in this new pandemic setting.

The present review aims to encompass the currently hypothesized pathophysiological mechanisms of liver injury in patients with COVID-19 mediated by both the direct invasion of the virus and its indirect effects and analyze the consequence of the pandemic in patients with chronic liver disease and liver tumors, with particular regard to the management strategies that have been implemented to face this worldwide emergency and that can be further improved.

## 2. Direct Viral Damage

Current data suggest that liver injury in COVID-19 is most likely mediated by systemic inflammation and the consequent accentuated immune response rather than the direct cytopathic effect of viral infection of liver cells. Moreover, liver damage is likely also sustained by the common drug toxicity of antivirals and by the hemodynamic instability secondary to multiorgan dysfunction in severely ill patients [17]. However, several findings support the theory that liver injury in COVID-19 patients might be, at least partially, caused by a direct attack by the virus.

SARS-CoV-2 expresses a surface glycoprotein, called spike, on the viral envelope, which binds to the receptor angiotensin-converting enzyme 2 (ACE2) on human host cells and, thus, mediates viral entry into the host cell cytoplasm [18]. Transmembrane protease serine 2 (TMPRSS2) and the paired basic amino acid cleaving enzyme (FURIN) also play an important role in the infection process since they facilitate the priming of the spike protein [19]. After the virus has fused with the host-cell membranes, the translation of the replicase enzyme from the virion genomic RNA occurs and the encapsidation results in the formation of the mature virus after replication and the subgenomic RNA synthesis. Virions are then transported in vesicles to the cell surface and released through exocytosis [20].

In the pre-pandemic era, ACE2 has been already recognized as the receptor for cell entry for other coronaviruses, including SARS-CoV, which was responsible for the Severe Acute Respiratory Syndrome (SARS) outbreak in 2003 [21,22]. In contrast, MERS-CoV was found to utilize dipeptidyl peptidase-4 (DPP-4) rather than ACE2 as its functional receptor for establishing infection in cells and causing Middle East Respiratory Syndrome (MERS) [23,24]. 

All three human host receptors (ACE2, TMPRSS2, and FURIN) involved in COVID-19 entry are variably expressed in liver tissues. However, the expression level of the ACE2 receptor was found to be 20 times higher in cholangiocytes compared to hepatocytes, suggesting that hepatocytes might not be targeted by the virus, or at least not through ACE2 [25,26]. For this reason, several authors have postulated that liver damage in patients infected by COVID-19 might be primarily due to the impairment of the barrier and the bile acid transporting functions of cholangiocytes, with consequent bile acid accumulation and hepatobiliary damage [27].

However, it has been recently demonstrated that hypoxia, systemic inflammation, and pre-existing liver disease could induce ACE2 and TMPRSS2 expression in hepatocytes, thus potentiating viral hepatotropism and increasing hepatic susceptibility to SARS-CoV-2 infection [28,29,30]. Moreover, in vitro experiments showed that the spike protein significantly increases the affinity for its receptor when it is pre-incubated with trypsin [18]. Since liver epithelial cells express trypsin and a plethora of other serine-proteases, which constantly remodel the extracellular matrix, ACE2 expression required for SARS-CoV-2 target and recognition in the liver might be lower than in other tissues with reduced extracellular proteolytic activity [31]. Similarly, it has been recently discovered that the spike protein of SARS-CoV-2 has a FURIN-like proteolytic site never observed in other coronaviruses of the same lineage [32]. Interestingly, FURIN is predominantly expressed in organs that have been proposed as permissive for SARS-CoV-2 infection, including the liver [25].

Whilst the tissue-specific factors controlling SARS-CoV-2 infection are still poorly understood, there is increasing recognition of the role of additional accessory receptors in viral entry. In this context, it is worth considering a recent study that proposed high-density lipoprotein scavenger receptor class B member 1 (SRB1) as a cell entry facilitator for SARS-CoV-2. SRB1 acts as a critical receptor that affects HCV entry, an infection that shares some molecular features with SARS-CoV-2 liver tropism. It is plausible that even if the expression of ACE2 in hepatocytes is relatively low, high levels of SRB1 may enhance the potential for SARS-CoV-2 entry in these cells [33].

In support of direct cytopathic damage of SARS-CoV-2, there are reports on a limited number of liver biopsy samples from COVID-19 patients that demonstrated moderate microvesicular steatosis, slightly watery degeneration or necrosis of hepatocytes, mild sinusoidal dilatation, and lymphocytic endotheliitis [34,35]. Similarly, a recent study with electron microscopy of two postmortem liver samples identified spiked, ‘corona-like’ inclusions in the cytoplasm of hepatocytes of COVID-19 patients with hepatic dysfunction, together with mitochondrial swelling, endoplasmic reticulum dilatation, and cell membrane dysfunction [36]. However, no immunolabeling of these particles was performed, raising concerns over the specificity of these particles in postmortem autolyzed samples, since the same observed changes may be seen also during multi-organ dysfunction associated with critical illness, drug-induced liver injury, and fatty liver disease [37,38]. Finally, although in some studies viral RNA could be detected in a substantial number of liver samples [33,39], this result could not be confirmed by others [40]. Moreover, despite viral RNA being recently found by PCR in the liver of 55% of COVID-19 patients [35], there was no correlation between PCR positivity and any of the histologic changes. Therefore, even if some studies have documented traces or evidence of the direct invasion of the virus in liver cells, further studies are needed to confirm a COVID-19-related cytopathic injury of the liver.

## 3. Indirect Viral Damage

Most of the available data demonstrate that hepatic injury during SARS-CoV-2 infection is more likely sustained by systemic inflammation and the consequent immune dysregulation [15]. In fact, due to its unique anatomical location, the liver is highly exposed to circulating antigens, endotoxins, inflammatory signals, and viral particles, which reach the liver either from the systemic circulation via arterial blood or the gastrointestinal tract through the portal vein. In particular, the overproduction of pro-inflammatory cytokines (such as IL-6 and IL-1) by monocytes and the dysregulation of lymphocytes with lymphopenia (including CD3+, CD4+, and CD8+ T cells) have been demonstrated to lead to harmful tissue damage, both locally and systemically, thus also causing liver damage [41,42]. For example, IL-6 plays a pivotal role in the pathophysiology of cytokine-driven hyperinflammatory COVID-19 status, and its serum levels have been strongly correlated with elevated transaminases levels in hospitalized patients with COVID-19 [43,44]. As already observed in other systemic viral infections [45], in fact, IL-6 can directly activate the hepatic immune cells and induce a transient elevation of aminotransaminases, a phenomenon called “bystander hepatitis”.

Moreover, the hyperinflammatory status of COVID-19 patients can induce disseminated intravascular coagulation, resulting in multiple organ failures and death in severe cases [46,47]. The complex vascularization of the liver makes it particularly exposed to circulatory alterations, therefore COVID-19-associated liver damage might be partially caused by this abnormal coagulation process and the consequent ischemic and hypoxic damage by microvascular thrombosis [48]. After COVID-19 infection, in fact, endothelial cells become inflamed and activated by the excessive release of cytokines derived from the exaggerated systemic immune response. This process leads to the upregulation of a wide range of pro-inflammatory cytokines, chemokines, and adhesion molecules, which promotes the recruitment of leukocytes, causes endothelial dysfunction, and produces vascular permeability and edema [49]. In confirmation of this, liver biopsies from COVID-19 patients reported massive dilation of portal vein branches, marked derangement of intrahepatic blood vessels, intravascular thrombosis, fibrin microthrombi and endothelitis in liver sinusoids, and hepatocytes necrosis [50,51,52]. In addition, some evidence suggests that endothelial cell activation might be also directly induced by SARS-CoV2 particles, due to the high levels of ACE2 and TMPRSS2 expressed by the endothelium [53] Finally, through its binding to ACE2, COVID-19 seems to induce both the overstimulation and imbalance of the Renin-Angiotensin-Aldosterone System, which can further contribute to vessel contraction and pro-hypertrophic and pro-fibrotic responses of endothelial cells [54].

Respiratory insufficiency and hypotension in the setting of COVID-19-related ARDS might contribute to hepatic hypoxemia and hemodynamic alterations, are likely also potentiated by reperfusion injury, and further aggravate the metabolic disturbances and liver function [55,56]. Under the conditions of ischemia and hypoxia, in fact, lipid accumulation, glycogen consumption, and adenosine triphosphate depletion of hepatocytes can inhibit cell survival signal transduction, rapidly leading to hepatocyte death. Moreover, the oxidative stress response promotes the continuous increase in levels of reactive oxygen species, with the activation of transcription factors and the further release of various pro-inflammatory factors that induce liver damage [57].

Finally, gut dysbiosis may indirectly contribute to liver damage through the translocation of endotoxins and bacteria with consequent activation of Kupffer cells, macrophages, and dendritic cells and the production of high levels of proinflammatory cytokines. In fact, the infection of intestinal cells by the SARS-CoV-2 virus can directly aggravate pre-existing gut dysbiosis and increase intestinal permeability, facilitating microbial metabolites transfer to the liver by the portal vein [58].

## 4. Drug-Induced Liver Injury

The liver is the leading site for metabolism and elimination for many drugs, including many of those being administrated for COVID-19 treatments. Therefore, not surprisingly, drug-induced liver injury (DILI) is more often described in patients with COVID-19 compared to routine practice. In particular, a recent meta-analysis showed a pooled incidence of DILI in COVID-19 patients of 25.4%, significantly higher than in the general population [59]. In fact, severe and critically ill COVID-19 patients are often treated with multiple drugs, including antipyretic non-steroidal anti-inflammatory drugs, antibiotics, immune modulators (such as tocilizumab and hydroxychloroquine), and antivirals (including lopinavir, ritonavir, and remdesivir), which all have the potential for liver injury. Moreover, the treatment of COVID-19 has seen the use of many therapeutic options, including the repurposing of older drugs, as well as the administration of off-label drugs or dosages not routinely used. Furthermore, the discontinuation of treatments in patients with underlying liver diseases or the potential interaction of these medications with the abovementioned COVID-19 treatment drugs might aggravate the risk of liver injury [60].

The reported histopathological changes described in some of the COVID-19 patients with liver damage are consistent with DILI. In particular, bioptic specimens show features compatible with a mixed injury pattern, i.e., with both hepatocellular and cholestatic damage, characterized by hepatocyte necrosis and inflammation along with bile stasis [61]. 

Due to their wide use, DILI in COVID-19 patients has been mostly reported with the use of antivirals [62]. For example, remdesivir has been associated with liver injury in early trials, causing mild to moderate elevations in transaminases within 5 days of therapy in 10–50% of patients [63]. Similarly, DILI has been shown in approximately 37% of patients in treatment with lopinavir/ritonavir [59]. Therefore, antiviral treatment should be used with caution, considering both the effectiveness and potential adverse effects [64].

During the early days of the pandemic, both hydroxychloroquine and chloroquine were widely explored for the treatment of COVID-19 infections due to their immunomodulator effect and proposed role in decreasing hyperinflammatory status. Hydroxychloroquine therapy has not been associated with ALT abnormalities and is an extremely rare cause of clinically apparent acute liver injury. However, hydroxychloroquine and chloroquine are both metabolized in the liver and their metabolites may accumulate in this organ, thus caution should be used in patients with liver cirrhosis [65,66].

Monoclonal antibodies such as tocilizumab (an anti-IL-6 receptor) have been indicated to counteract the cytokine storm associated with severe disease. These drugs have been associated with a transient elevation in the hepatic enzymes, but cases of severe liver injury have rarely occurred. Nonetheless, patients with a pre-existing hepatitis B viral infection have experienced reactivation of the virus after treatment with tocilizumab, thus their usage should be approached very carefully [67].

Similarly, it should be acknowledged that prolonged use of corticosteroid therapy can cause hepatic steatosis and increase the risk of developing reactivation of latent infections, such as viral hepatitis B. [7,63,68].

Non-steroidal anti-inflammatory drugs such as acetylsalicylic acid are one of the most common causes of drug-induced liver injury in the general population. With fever being a symptom of COVID-19, antipyretics have been highly prescribed by physicians during the pandemic, often off-label in those patients who did not require hospitalization, thus their role in drug-induced liver damage should not be underestimated [69]. Paracetamol has both antipyretic and analgesic effects and has been widely used as an adjunct therapy, despite being a well-known cause of dose-dependent DILI. In particular, the underlying mechanism for paracetamol toxicity is its direct damage through a metabolite generated from liver metabolism [65].

Finally, antibiotic consumption is high among COVID-19 patients, especially in hospitalized settings where the main rate of usage reaches up to 74% [70]. In addition to possibly increasing antimicrobial resistance [71], the inappropriate use of antibiotics, especially those with well-known hepatotoxicity (such as azithromycin), may lead to additional liver damage [72].

## 5. Hepatic Manifestations in COVID-19 Patients

Currently, there is no standardized definition of COVID-19-related liver injury, and the diagnostic time point (admission or during disease progression) is not always reported [73]. In fact, despite some researchers having defined liver injury in COVID-19 patients as any liver function parameter above the upper limit of normal [74,75], others have introduced different threshold values (an increase in liver enzymes higher than 2 or 3 times the normal values) [13] and even further classified different liver injury patterns (hepatocellular type, cholangiocytes type, and mixed type) [14,76]. Because of the different criteria considered, researchers may have overestimated or underestimated COVID-19-related liver damage, partially explaining the mixed results so far and jeopardizing the generalizability of the conclusions and the practical clinical implications derived from these studies. Therefore, an international consensus in this regard is urgently needed.

Abnormal liver function tests in patients with COVID-19 were first reported in a cohort of 99 patients in Wuhan [77]. In particular, nearly all patients (98%) presented with decreased albumin levels, whereas 28% and 35% of them showed a moderate increase in alanine aminotransferase (ALT) and aspartate aminotransferase (AST) levels, respectively. In the countless following studies, abnormal liver function tests were commonly reported in COVID-19 patients [78], mainly in the form of a hepatocellular injury pattern rather than cholestatic [62,79]. In fact, a transient increase in aminotransferases is frequent in patients with COVID-19, along with an elevation of lactate dehydrogenase levels, followed by a significant decrease in albumin later in the course of the disease [80,81]; in contrast, alkaline phosphatase and gamma-glutamyl transferase, representatives of bile duct injury, do not increase significantly, and jaundice is uncommon [15,48,75]. In particular, the elevation of AST levels seems to be more frequent and significant than the increase in ALT [82].

The incidence of these abnormal liver tests is reported to be significantly higher in patients with severe COVID-19 compared with those with mild disease [83,84]. For example, in a recent meta-analysis, aminotransferases elevation was observed in 23% of the patients with mild symptoms, whereas hypoalbuminemia was observed in 61%, percentages that increased to 40% and 76%, respectively, in severe patients [6]. Similarly, increased AST was observed in 62% of patients with COVID-19 in the intensive care unit (ICU) compared to 25% in non-ICU patients [15]. 

Other studies have demonstrated that liver enzyme levels could be used as important clinical markers, since their elevation is associated with a higher risk of in-hospital mortality, a longer hospital stay, and other adverse clinical outcomes, such as the need for vasopressor drugs and mechanical ventilation [82,85,86]. Underlying the important mutual relation between the liver and COVID-19 disease, a recent study demonstrated that liver test abnormalities upon hospital admission, in particular, elevated ALT or AST, can be used to predict the severity of COVID-19 [13]. Therefore, even if non-specific and equally reported also in non-COVID patients [6], these parameters should be always monitored during hospitalization due to their prognostic value.

COVID-19-related liver injury is usually mild and transient, and liver failure is exceedingly rare, being reported only as anecdotical case reports in the setting of a severe disease with sepsis and coagulopathy requiring the administration of multiple drugs. Therefore, whether this drastic derangement of liver function is secondary to true viral damage rather than a bystander to the multiorgan pathophysiology of critical illness or rather the result of the hepatotoxic potential of high doses of antiviral drugs requires further discussion [87,88].

Finally, as previously stated, in addition to systemic inflammation, liver dysfunction could be partially mediated by COVID-19-related coagulopathy and intrahepatic microvascular thrombosis [89,90,91]. This theory is also supported by laboratory findings, as demonstrated by a recent study that found an independent association between higher D-dimer levels and elevation of ALT [92]. Similarly, the abnormally high levels of transaminases detected in COVID-19 patients whose liver samples were analyzed post-mortem showed aspects of intravascular thrombosis, further suggesting possible liver damage linked to impaired coagulation [52]. 

## 6. The Role of Imaging in COVID-19-Related Liver Injury

During the early phases of the pandemic, when viral tests were not available or were scarce, imaging provided considerable help in the diagnosis of COVID-19 pneumonia [93]. At present, following the advancement in diagnostic laboratory techniques, its role has evolved, and now imaging is pivotal to the detection and monitoring of COVID-19 complications, both pulmonary and visceral, the evolution of which may be heterogeneous and unpredictable [94,95,96].

To date, no specific hepatic imaging findings related to COVID-19 have been reported in the medical literature [97]. However, some studies have reported non-specific changes in the appearance of the liver on imaging following COVID-19 infection, namely features of hepatic steatosis, suggesting a possible association between these two conditions [98]. On ultrasound (US), in patients after COVID-19, the liver may appear “brighter” due to an increase in echogenicity compared to the renal cortex or spleen, with the loss of physiological hyperechogenicity of the portal branches walls and posterior attenuation of the ultrasonic beam and the consequent failure to visualize the diaphragm [99]. These changes have been also documented in a recent study with Multiparametric US (mpUS), where the sonographic evaluation of liver parenchyma in individuals after COVID-19 revealed significantly more frequent liver steatosis compared to the clinically healthy control group but also increased stiffness (fibrosis) and viscosity (inflammation) values, both indicative of liver injury. However, despite these mpUS features being correlated with an increase in biochemical markers of liver injury, they were not associated with correspondent CT or MR findings [100]. On the other hand, several studies have reported a considerably greater frequency of hepatic steatosis by Computed Tomography (CT) scans in confirmed COVID-19 cases compared to controls, defined as a hepatic attenuation of at least 10 Hounsfield Units (HU) compared to the density of the spleen or absolute liver attenuation of less than 40 HU [101,102]. Moreover, hepatic steatosis resulted more prevalent in patients with laboratory liver dysfunction (approximately 19% vs. 8%) and in those with severe pneumonia (approximately 73% vs. 27%), being similarly reported in patients with and without fatty liver disease [103,104,105]. Furthermore, it seems that hepatic steatosis is only a transient change in COVID-19 patients, as it has been demonstrated to gradually recover on follow-up CT scans, either due to the concomitant improvement of the viral infection or the development of hepatic fibrosis [105].

Currently, only one study has evaluated the Magnetic Resonance Imaging (MRI) liver features following COVID-19 infection [106]. According to the authors, the patients who recovered from COVID-19 presented a lower hepatic T1 relaxation time in the 3-month follow-up compared with the control group of normal volunteers, suggesting that this result might be attributed either to hepatic steatosis and/or iron deposition secondary to hepatocyte impairment and reduce the production of hepcidin. Moreover, higher diffusion/perfusion parameters were also found in the 3-month follow-up group of recovered COVID-19 patients, which may be the result of the inflammation-induced interstitial edema and focal hepatic necrosis. During the 1-year follow-up, all these MRI findings showed mild recovery, indicating that the liver impairment after infection was relieved over time via liver regeneration.

In any case, all these hepatic imaging alterations are highly non-specific, more likely representing the “collateral” consequences of a multisystemic hyperinflammation process and/or the result of multidrug treatments rather than evidence of a direct SARS-CoV-2 invasion of the liver.

## 7. COVID-19 in Patients with Pre-Existing Chronic Liver Disease

Patients with chronic liver disease are frequently immunocompromised and, as already stated, may exhibit an increased expression of ACE2 in hepatocytes as an injury response to liver fibrosis [30]. Therefore, theoretically, these patients might be more susceptible to SARS-CoV-2 infection [28,107,108]. Nonetheless, based on currently available evidence, patients with chronic liver disease do not appear to be at a higher risk of infection compared to other individuals in the general population [56,109,110].

However, the risk of mortality from COVID-19 was significantly increased in these patients (with a risk ratio of 3) [111], as also confirmed in a large survey including more than 17 million cases [107], especially in those with cirrhosis (where the risk ratio reaches up to 4.6), even in the absence of respiratory symptoms at the time of diagnosis [112,113,114].

In addition to this direct effect, given the great burden of health system reorganization and lockdown strategies, the care of patients with pre-existing chronic liver disease has been widely disrupted during the pandemic, leading to failure of the screening and follow-up, a higher prevalence of related complications (including variceal bleeding, hepatic encephalopathy, liver failure, and hepatocellular carcinoma development), and delayed access to treatments and liver transplantation [16,115]. On top of that, social distancing has further induced and aggravated alcohol use disorders, facilitating the relapse of alcohol addiction, while also favoring the consumption of processed food and the adoption of a sedentary lifestyle, which are well-known risk factors for morbidity and mortality (Table 1) [17,116]. The future long-term consequence of these modifications and delays in management and treatment options are still to be realized and will presumably lead to a transformation of the curable into the incurable [117].

### 7.1. COVID-19 in Patients with Non-Alcoholic Fat Liver Disease (NAFLD)

Individuals with a poorer prognosis of COVID-19 are typically older (>60 years old) with metabolic co-morbidities such as obesity and diabetes, a profile that is similar to those at increased risk of NAFLD [111,118]. Therefore, it is not surprising that patients with NAFLD have a higher risk of progression to a severe form of COVID-19 (45% vs. 7%) and a likelihood of abnormal liver function tests from admission to discharge (70% vs. 11%) compared to those without NAFLD, even in younger patients without other comorbidities [119]. Moreover, this subgroup of patients tends to present a longer period of viral replication and dispersion [76,120]. 

However, recent observations suggest the association between NAFLD and severe COVID-19 disease might be confounded by other metabolic perturbations underlying NAFLD, such as obesity, diabetes, and hypertension, which are all well-known factors that increase the risk of all severe infections. In particular, it seems that obesity might represent the most important metabolic perturbation associated with severe COVID-19, and some researchers suggest that it is likely the only causal risk for progression to severe pneumonia, even after adjusting for co-morbidities including NAFLD [121]. As confirmation of this, despite obesity having also been associated with a greater risk of hospitalizations, mechanical ventilation, extensive coagulopathy, and death, the same does not apply to hepatic steatosis [122,123,124].

Similar to NAFLD, in fact, the adipose tissue of obese patients induces a low-grade chronic inflammation with increased serum levels of pro-inflammatory cytokines and is thought to express high levels of ACE2, thus potentially functioning as a SARS-CoV-2 reservoir with prolonged viral shedding time [125]. Moreover, there is no evidence of increased liver uptake of SARS-CoV-2 in NAFLD patients [126], whereas SARS-CoV-2 mRNA has been detected in adipocytes, especially those of visceral fat [127]. Therefore, despite the clear interlink between NAFLD and obesity, whether hepatic steatosis simply represents a concomitant metabolic dysfunction in obese COVID-19 patients or rather an active factor for pneumonia progression is still debated, and further evidence is needed.

Finally, it is fundamental to stress the sneaky and collateral effect of the pandemic on NAFLD patients, whose daily routine has been severely impacted by lockdowns and social isolation, with subsequently more sedentary and unhealthy lifestyles. Therefore, COVID-19 will presumably be an indirect cause of the expansion of the NAFLD epidemic in the next few years [128].

### 7.2. COVID-19 in Patients with Cirrhosis

Similar to other patients with pre-existing chronic liver disease, cirrhotic patients showed an increased risk of COVID-19-related liver injury and mortality [129,130]. Moreover, recent data appear to indicate a three-fold increased risk of death among cirrhotic patients compared to patients with other chronic liver diseases, and this increases with the stage of cirrhosis, the Child-Turcotte-Pugh class, or the Model for End-Stage Liver Disease (MELD) score [107,131,132].

The greater mortality of this subgroup of patients could be related to cirrhosis-associated immune dysfunction, which would lead to an aberrant inflammatory response during infection with the reduction of complement components, activation of macrophages, and impairment of lymphocyte and neutrophil function [133]. Moreover, COVID-19 cytokine activation may induce hepatocyte apoptosis and necrosis, which, in the setting of a diminished liver reserve, may trigger acute-on-chronic liver failure [134]. A large multi-center cohort, in fact, showed that hepatic decompensation was strongly associated with COVID-19 infection, increasing the risk of death from 26.2% to 63.2% [135]. Similarly, another study found that liver function in cirrhotic patients with COVID-19 upon hospital admission declined compared to the last visit before SARS-CoV-2 infection [113]. Interestingly, the mortality of cirrhotic patients with and without COVID-19 is reported to be similar, whereas the mortality of cirrhotic patients with COVID-19 is higher compared to those suffering from COVID-19 alone [136]. However, sarcopenia is a frequent complication in end-stage liver disease [137,138] and, since it has also been associated with poor clinical outcomes in COVID-19 patients [139], it could be interesting to evaluate whether its presence might influence both morbidity and mortality risk in cirrhotic patients since these data are lacking. Moreover, COVID-19 is now considered a risk factor for the onset and progression of sarcopenia, thus data regarding the relationship between cirrhosis and sarcopenia during the pandemic outbreak deserve to be collected and analyzed.

However, the main cause of death in most cirrhotic patients with COVID-19 is still represented by pulmonary disease rather than the progression of the underlying liver disease [133]. Therefore, it is plausible that pulmonary thromboembolism, a hallmark of critical COVID-19, has a contributory role given the additional hypercoagulable state associated with cirrhosis [140]. 

Finally, as testified by the drastic reduction in hospitalization for cirrhosis following the onset of the pandemic, delays in both follow-up and treatment have occurred, leading to more frequent and advanced cirrhosis-related complications. For example, acute variceal hemorrhage developed in patients who had not undergone timely endoscopic surveillance, whereas postponed routine elective paracentesis for tense ascites was converted into one requiring emergency hospitalization [141]. Therefore, the recognition that patients with cirrhosis are particularly vulnerable to the severe complications of COVID-19 is paramount. and consideration of early admission for these patients is encouraged. This careful attention should also be maintained following the availability of COVID-19 vaccines since recent data seem to suggest a decreased efficacy and durability of protection in patients with cirrhosis [142].

### 7.3. COVID-19 in Patients with Viral Hepatitis

Whether chronic viral hepatitis specifically affects the COVID-19 course remains unknown. Some researchers argue that there might be an immune response overlap between chronic viral hepatitis and COVID-19, in which a baseline inflammatory status could be further potentiated following SARS-CoV-2 infection [143].

Some studies suggest that patients with chronic hepatitis B are more vulnerable to COVID-19 [144] but others revealed that these patients do not appear to have a more severe course of the disease [145,146].

Similarly, little is known about the clinical course of COVID-19 in hepatitis C patients, but the current few studies available do not suggest an increased mortality or ICU admission rate compared to patients without HCV infection [147]. However, it recently emerged that patients with active HCV infection have a significantly higher risk of severe COVID-19 and increased mortality compared to the non-active HCV group [148]. 

Glucocorticoids or immunosuppression therapies are widely used in patients with severe COVID-19, therefore there is an urgent need for data from various populations to inform the risk stratification and management of those who are at elevated risk of HBV and HCV reactivation [149]. A recent systematic review, however, demonstrated a small risk of HBV reactivation overall following Tocilizumab administration (approximately 3%) and an even lower risk for HCV flare [150].

Nonetheless, what appears certain is that, as a consequence of this pandemic, most viral hepatitis elimination programs have diminished or stopped altogether [130]. Recent estimates have calculated that a 1-year delay in viral hepatitis diagnosis and treatment will result in an additional 44,800 hepatocellular carcinoma cases and 72,300 deaths from HCV globally in the next 10 years, thus the collateral burden of the COVID-19 pandemic on patients with viral hepatitis will become clear in the next few years [151].

### 7.4. COVID-19 in Patients with Alcohol-Associated Liver Disease (ALD)

The consumption of alcohol has increased in the present pandemic due to various reasons such as social isolation, job insecurity, and higher anxiety and depression [152]. Furthermore, the unavailability of professional help and support groups such as Alcoholics Anonymous may have led to psychological decompensation, alcohol abuse, and relapse of alcohol addiction [128]. This trend, coupled with the ubiquitous availability of inexpensive alcohol, determined a worldwide increase in alcoholic beverages. For example, as stay-at-home orders began, alcohol sales increased by up to 55% in the United States during March 2020, along with an alarming 262% increase in online sales compared to 2019 [153]. Similarly, in Canada, there was a 38% relative increase in monthly alcohol sales in March 2020 compared with March 2019 [154]. The same results were reported also in China and France, where 32% and 25% of regular alcohol drinkers reported an increase in their usage amount during the pandemic, respectively [155,156].

The increased consumption of alcohol has inevitably induced and aggravated alcohol-related disorders, in particular alcohol-associated liver disorder (ALD), which may predispose individuals to worse outcomes from COVID-19 [17]. In fact, despite the fact that there are very limited data on the effect of COVID-19 in patients with ALD, the few studies currently available showed that these patients had an increased risk of COVID-19-related complications. The first reason for this is that alcohol use and ALD disrupt the innate and adaptive immune systems by affecting the survival and function of immune cells important in mounting a defense against viral infections. Second, patients with ALD often have other comorbidities, including obesity, diabetes mellitus, chronic kidney disease, and tobacco use, which have been all independently associated with severe COVID-19 outcomes [152,154]. Additionally, a recent study, after adjusting for potential confounders such as sex, disease severity, and other comorbidities, reported that patients with ALD have a higher risk of mortality for COVID-19 even compared to patients with cirrhosis due to other etiologies [129].

On top of that, an interesting study found evidence of a substantial and rising burden of ALD, particularly regarding liver transplantation [157]. In fact, during the COVID-19 era, the percentage of ALD on the liver transplant waiting list significantly increased (approximately 3%), surpassing that of HCV and NASH combined for the first time.

In consideration of this, clinicians and hepatologists should be carefully focused on the prevention of infection by following strict epidemiologic recommendations, as well as on the prevention of alcohol abuse and relapse of alcohol addiction by implementing vigorous social and psychiatric care via alternative telephone or online resources. 

### 7.5. COVID-19 in Patients with Autoimmune Hepatitis (AIH)

Regarding patients with AIH, it has been hypothesized that their immunosuppressive state can predispose them to SARS-CoV-2 infection and a greater risk for severe disease. In particular, the SARS-CoV-2 virus may trigger autoimmune mechanisms in genetically predisposed subjects, hyperstimulating the immune system and exposing foreign peptides homologous to human peptides (molecular mimicry), thus leading to the development of autoantibodies [158]. However, there are scarce reports of AIH triggered by COVID-19 infection, and several confounding factors were always present [159].

Most of the current studies agree that AIH patients under immunosuppression, despite the theoretically suspected susceptibility, have the same prevalence of COVID-19 infection compared to the general population [160,161,162] and a similar disease course [163,164]. Therefore, there is growing evidence suggesting that immunosuppression may provide some protection from lung damage in patients with COVID-19 or at least counterbalance COVID-19-driven hyperinflammatory status.

Since the outcome was favorable in most cases, the most important association for the study of the liver currently advises against reducing immunosuppressive therapy in these patients, except in special circumstances such as severe COVID-19, bacterial/fungal superinfections, or lymphopenia [165,166].

### 7.6. COVID-19 in Patients with Hepatocellular Carcinoma (HCC)

Patients with both COVID-19 and a history of cancer are more vulnerable to severe disease and have an increased chance of mortality, as well as ICU admission, especially if they have received chemotherapy within 1 month [167]. Since most patients with Hepatocellular Carcinoma (HCC) have pre-existing chronic liver disease, it is possible to assume that this subgroup of patients is even more susceptible to the effects of COVID-19 compared to other patients with cancers.

The COVID-19 pandemic has had a tremendous impact on both the management and treatment of patients with HCC, causing the rescheduling of screening exams and the shift of liver cancer therapy toward nonsurgical procedures. 

Several studies reported a significant decrease in the number of new HCCs diagnosed (up to 37%) during the pandemic, which is in marked contrast to the steady increases in previous years and suggests that incident cases went undiagnosed during this period [168,169]. In addition, HCCs diagnosed following the COVID-19 outbreak were significantly larger and more frequently associated with lymphadenopathy and extrahepatic disease, with a consequent shift toward a higher tumor burden at diagnosis. Similarly, symptomatic presentation for HCC moved from being the least common mode of presentation to the most common, accounting for approximately 40% of all cases (vs. 24% in the pre-pandemic era), as demonstrated by the high rate of spontaneous hemorrhages. Furthermore, tumor progression occurred more rapidly and aggressively in HCC patients during the pandemic due to delayed or discontinued follow-up and treatment [170]. To overcome the restrictions in diagnostic procedures due to reduced radiologic capacity during the pandemic, both the American and the European association for the study of the liver recommend continuing surveillance imaging of HCC patients and high-risk cirrhotic patients with a reasonable delay of a maximum of 2–3 months, which is considered safe, possibly prioritizing those at increased risk if resources are limited. Moreover, the increasingly widespread possibility to revise diagnostic images and conduct multidisciplinary meetings online, also known as telemedicine, should be encouraged [165,171].

Elective hospital admissions were also delayed at the peak of the COVID-19 pandemic due to low hospital capacity and the fear of viral spread. As recently reported, this led to delayed treatment in a significant percentage of HCC patients (21–26%) for 1–2 months or longer [172,173]. Despite the fact that indications of surgery, locoregional therapy, and liver transplantation in HCC patients have not changed following the COVID-19 outbreak, a significant difference in the modification of the treatment strategy was noted. In particular, surgical resection and liver transplantation were the most affected due to a shortage of anesthetists and other medical personnel, the shortage of ICU beds, and a decline in the number of donors [174]. Therefore, locoregional procedures such as transarterial chemoembolization (TACE) and radiofrequency ablation became the most important tools in treating patients with HCC, as they acted as a salvage procedure to decrease the risk of cancer progression during the waiting period. Moreover, in addition to a good outcome in oncological terms, nonsurgical treatments are characterized by a lower complication rate and shorter hospital stay, which reduce the patient’s risk of contracting the virus during hospitalization [175,176]. 

Systemic therapy in advanced HCC should be maintained according to guidelines, and sorafenib administration should be continued without any change in its dose. Conversely, immune-checkpoint inhibitor therapy in HCC patients with severe COVID-19 should be temporarily withdrawn, whereas the decision on whether to continue or reduce its dose in non-severe patients should be taken on a case-by-case basis. Moreover, it is generally recommended to reduce the exposure of patients at the infusion center, which represents a hotspot for COVID-19 infection, through the implementation of home blood sampling and drug delivery, as well as video calls to manage common adverse events [177,178].

In conclusion, the pandemic has led to a rapid and prompt reorganization of activities in order to minimize its effect on patient outcomes and reduce the risk of exposure to SARS-CoV-2 as much as possible. In particular, the adoption of telemedicine together with the optimization of protocols and the implementation of both intra- and postprocedural workflows seems to be the right path to follow [179].

### 7.7. COVID-19 and Liver Transplantation

The COVID-19 pandemic has substantially impacted solid organ transplantation, including the temporary inactivation of waitlist candidates who tested positive for SARS-CoV-2 infection with a consequent suspension of transplant activity, except for extremely super-urgent cases [180,181]. The first pandemic wave caused a reduction in organ donation compared to the same period in the previous year, as demonstrated by the 25% decrease in liver transplants reported by the United Network for Organ Sharing (UNOS) [182] and also later confirmed in a large population-based study across 22 countries worldwide, which showed an overall reduction in liver transplants of 11% [183]. Moreover, in most countries, transplantation activity only slowly recovered in the following months, due to the occurrence of subsequent pandemic waves [184].

Early studies, in fact, suggested increased perioperative mortality and morbidity in patients with COVID-19 undergoing an elective and emergent general surgery procedure, leading transplant centers to defer liver transplantation in patients with active COVID-19 infection [185,186,187]. Moreover, a further challenge was related to the staff involved in the surgical procedures, who cannot be easily replaced if testing positive [128]. Therefore, both European and American Liver Associations advised prioritizing patients who have a poor short-term prognosis, considering living-donor transplantations on a case-by-case basis, thus leading to an inevitable reduction of transplant activity [164,171].

Nonetheless, more recent studies suggested that liver transplant recipients with COVID-19 infection have a similar risk of mortality compared to the general population and that transplantation is not independently associated with hospitalization or death from COVID-19 [188,189]. An interesting study even surprisingly reported a lower mortality rate in liver transplant patients diagnosed with COVID-19 compared to the complementary general population [189]. This evidence led to the hypothesis that immunosuppression might have favorable effects in transplanted patients by disabling cytokine release syndrome and thus protecting these patients from hyperinflammation and ARDS development in COVID-19 [190,191,192]. Additionally, waitlist mortality among those with end-stage liver disease remains high, and it is unknown whether the risk of COVID-related mortality after transplant outweighs the risk of waitlist mortality [188].

Furthermore, clinicians agree with delaying liver transplantation wherever possible, depending on the local availability of resources such as ICU beds, ventilators, and blood donations. After transplantation, they discourage the reduction of immunosuppressive therapy, except in severe cases of bacterial or fungal infections superadded to COVID-19, or associated lymphopenia [165,171].

Currently, the use of liver grafts from donors testing positive for COVID-19 is still contraindicated, therefore screening of both donors (living and deceased) and recipients listed for liver transplantation is required [165,171]. To date, in fact, little is known about the transmission of the virus from donor to recipient. Moreover, liver transplant recipients are in an immune-compromised state, which puts them at higher risk of contracting COVID-19 infection and being a source of dissemination of infection to others (super-spreaders) [193]. 

However, recent studies found that recipient outcomes with the use of SARS-CoV-2-positive donor organs are similar to outcomes with negative testing, and patients receiving grafts from COVID-19-positive donors showed equivalent survival to those receiving grafts from COVID-19-negative donors [194,195]. Moreover, to date, there have been no reports of recognized transmissions of SARS-CoV-2 following blood transfusion, even in immunosuppressed patients [196,197]. Therefore, although, the number of transplants reported from positive SARS-CoV-2 donors is still relatively small, there is no clear evidence of additional risk to allograft survival [198]. 

Nonetheless, despite transplantation from donors with COVID-19 appearing feasible, especially in selected patients with imminently life-threatening organ failure, evidence is still lacking, and further studies are needed [197,198,199]. Furthermore, the impossibility of clearly defining the risk of transmission prevents the recipient from providing adequate informed consent and the exposure of healthcare workers during organ procurement remains a concern [184].

There have also been few reports of successful liver transplantation in individuals with asymptomatic or mild symptomatic COVID-19 infection without developing postoperative COVID-19 symptoms [200,201,202,203]. However, the decision to proceed with transplantation in this high-risk population must be balanced by the urgency of the procedure against potential COVID-19-associated perioperative risks; therefore, ideally, transplantation should always be deferred until acute COVID-19 infection is resolved.

## 8. COVID-19 Vaccination and the Liver

Following the development and widespread distribution of COVID-19 vaccines across the globe, sporadic cases of thrombotic events in young women after the vaccination with the Oxford-AstraZeneca vaccine have been reported [204], as well as anecdotal cases of different types of autoimmunity triggered by vaccines, including subacute cutaneous lupus erythematosus and Graves’ disease [205,206].

Similarly, rare case reports of hepatic complications mimicking autoimmune hepatitis (AIH) have been described in recipients of various vaccines, including the adenovirus-based vaccine of Oxford-AstraZeneca and the mRNA vaccines of Pfizer-BioNTech and Moderna, suggesting that these adverse events may be independent of the vaccine mechanisms [207,208,209,210,211]. Only association, but not causality, was suggested between COVID-19 vaccination and AIH due to the presence of other confounding factors, including other autoimmune diseases or concomitant treatment with drugs linked to the induction of autoimmune conditions. The majority of these cases were described in females and had a short latency time from vaccination to the onset of symptoms, which are still generally mild. Moreover, patients were treated with immunosuppressive therapy with remission within 5–6 months.

Despite the current data appearing to suggest a potential role of vaccines in unmasking AIH in predisposed individuals, whether there exists a causal relationship between COVID-19 vaccination and the development of autoimmune hepatitis remains to be determined. Although new variants of SARS-CoV-2 are emerging, vaccination remains an effective and reliable way to reduce disease severity, hospitalization requirement, and more importantly, mortality. 

## 9. The Impact of COVID-19 on Medical Personnel and Trainees

Burnout, defined as a multifaceted construct characterized by emotional exhaustion, depersonalization, and a low sense of personal achievement, was frequent among hepatologists and gastroenterologists following the COVID-19 outbreak [212]. During the pandemic, in fact, medical workers had to care for large numbers of infectious patients with a poorly understood disease, thus being afraid of getting infected and infecting their loved ones. Moreover, they were also often separated from colleagues, and many imposed self-isolation periods away from their families [213,214]. 

Additionally, the COVID-19 pandemic also had a disruptive effect on the training of hepatologists, gastroenterologists, radiologists, and other medical personnel, due to the redeployment and cancellation of educational activities. Following the pandemic outbreak, in fact, elective procedures were drastically interrupted leading to a significant gap in medical education and decreased physical and emotional well-being of trainees [180,215]. Moreover, global shortages of personal protective equipment and the attempt to minimize the risk of infection led to restricting staffing for procedures and excluding “non-essential persons” as fellows. The small outpatient residual volume during the COVID-19 outbreak and the frequent unavailability of training mentors who had tested positive for SARS-CoV-2 further contributed to an increase in trainees concerning attaining the program requirements and maintaining their procedural skills [180]. According to a recent European survey, in fact, the majority (84.5%) of gastroenterology trainees reported a high impact on training activities by COVID-19, whose residual activities mainly concerned urgent endoscopies and oncologic patients [216]. On top of that, approximately 30–50% of the gastroenterologist fellows were redeployed to COVID-19 wards, however, often without adequate training, thus further limiting their education [216,217].

To overcome these difficulties, the implementation of virtual training, with webinars held by specialist scientific societies and simulation-based training, should be promoted to meet trainees’ needs [215]. Moreover, the creation of a clear and open channel of communication between the program director and the fellows via mail and/or an active social media chat group, as well as psychological support, has already been proven to be useful [213]. Despite some studies proposing an extension of the training period or the relocation of trainees to COVID-free hospitals [218], it is important to acknowledge that the COVID-19 pandemic has permanently impacted our practice of medicine, thus the disease will likely still play a massive role in Hospital wards in the next years. Therefore, as the world begins to transition to a “new normal”, healthcare must find alternative and valid solutions to ensure continuity in providing patient care, possibly with an increase in medical support staff and the re-distribution of available resources, in order to address the increasingly frequent infectious threats. Moreover, similarly to other pandemics, COVID-19 tends to come in several waves over a protracted period until hopefully subsiding through vaccination and herd immunity. During the interpeak phases, every effort should be made to restore clinical liver activity to regular rates and indications to prevent accumulated deaths in the long term.

In this regard, a rapid increase in the use of telemedicine has occurred since the onset of the COVID-19 pandemic, with the continuous expansion of both access and coverage of its services [219]. Both European and American guidelines have already recommended the use of telemedicine to replace clinic visits, as well as replacing multidisciplinary team meetings to decrease the risk of the spread of COVID-19 [165,171]. Moreover, it also allows overcoming barriers to access by reducing or eliminating travel distance, frequency, and associated costs, as testified by patients themselves [219]. Recent studies, for example, have shown that telemedicine may be an effective way to remotely evaluate HCC patients for liver transplantation [220,221]. Similarly, virtual multidisciplinary tumor boards have been shown to be effective in completing HCC treatment evaluations in a shorter amount of time [222]. However, telemedicine still has several limitations, especially in developing countries, due to the restricted access to technology, the cost of implementation, and the training of staff members [152]. Furthermore, it is inappropriate to visit when a physical examination is crucial, exposing clinicians to legal liabilities [223].

## 10. Conclusions

The COVID-19 pandemic has profoundly strained healthcare resources around the world, and the liver is a secondary and often collateral target of this disease. In particular, despite the fact that there is mounting evidence supporting a cytopathic effect of viral infection of liver cells, liver damage seems to be mainly sustained by indirect effects of SARS-CoV-2, such as systemic inflammation with cytokine storm, hemodynamic instability, the alteration of the gut–microbiota axis, and drug-related toxicity.

Whatever the pathophysiological mechanisms behind the liver injury, the involvement of this organ in patients with COVID-19 can lead to important consequences, both direct and indirect. The direct consequences of liver involvement mainly involve subjects with pre-existing chronic liver disease, especially cirrhotic patients and those with an alcohol-related disease, who can develop a high number of complications and present a higher risk of hepatic decompensation, as demonstrated by an increase in their mortality rates. The indirect consequences of COVID-19 in the liver are instead expressed by the difficulties in the management of patients with liver disease, especially in those suffering from decompensated cirrhosis and HCC, as well as in the execution of their follow-up and the availability of all therapeutic possibilities, as demonstrated by the greater tumor burden and the reduction in surgical treatments, including liver transplants. Interestingly, patients who underwent liver transplantation and those with autoimmune liver disease seem to be protected from COVID-19, with no significant differences in mortality compared to the general population; nevertheless, it is still necessary to monitor these patients since their follow-up could be significantly affected due to the reorganization of healthcare and the increase in the workload following the pandemic.

Liver imaging in COVID-19 patients is highly nonspecific and seems to further confirm that this organ rather represents the result of collateral damage from a severe and generalized disease. Nevertheless, imaging can still prove useful, especially in patients with underlying liver disease, since it can help identify the complications that derive from the infection.

Attention must also be used in the use of drugs, especially in patients with pre-existing liver disease, because they could worsen an already impaired hepatic function and often even overcome the damage caused by the virus itself.

Vaccines are still our best weapon to fight the virus, and despite the few reported cases of side effects, their benefits far outweigh their risks.

Finally, telemedicine constitutes a possible solution to both the physical distancing and the re-organizational difficulties arising from the pandemic, and its role will be increasingly important in the near future, thus every effort must be taken to make this implementation as efficient and broad as possible.

## Figures and Tables

**Table 1 ijms-24-01091-t001:** Effects of COVID-19 infection in patients with pre-existing liver disease.

Patients with NAFLD	Increased risk of severe disease Increased mortality Care disruption and delayed access to treatments More sedentary and unhealthy lifestyles due to social distancing
Patients with Cirrhosis	Greatly increased risk of severe disease Increased mortality Increased rate of complications (including variceal bleeding, hepatic encephalopathy, liver failure and HCC development) Care disruption and delayed access to treatments and liver transplantation
Patients with Viral Hepatitis	Increased risk of severe disease Increased mortality Care disruption and delayed access to treatments elevated risk of HBV and HCV reactivation secondary to COVID-19 treatments
Patients with ALD	Increased risk of severe disease Increased mortality Care disruption and delayed access to treatments Increased alcohol consumption and risk of abuse relapse due to social distancing
Patients with AIH	Probably lower susceptibility to infection and severe disease due to immunosuppressive therapies Care disruption and delayed access to treatments
Patients with HCC	Increased risk of severe disease Increased mortality Diagnostic delay and diagnosis at more advanced stages Care disruption and delayed access to surgical treatments and transplantation Increased performance loco-regional treatments for bridging and downstaging
Patients in list for LT or who underwent LT	Suspension of transplant activity due to candidates and staff tested positive for COVID-19 and/or unavailability of hospital beds Probably lower susceptibility to infection and severe disease due to immunosuppressive therapies

NAFLD: Non-Alcoholic Fat Liver Disease; ALD: Alcohol-associated Liver Disease; AIH: Autoimmune Liver Disease; HCC: Hepatocellular Carcinoma; LT: Liver Transplant.

## Data Availability

Not applicable.

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
