# Peer review of "COVID-19: Has the Liver Been Spared?"

_ijms, 2023, doi:10.3390/ijms24021091_

Round 1

Reviewer 1 Report

COVID-19: Could the liver have been spared? 

Nicolò Brandi, Daniele Spinelli, Alessandro Granito, Francesco Tovoli, Fabio Piscaglia, Rita Golfieri, Matteo Renzulli

1.The title is really striking and partly reflects the main subject of the manuscript.

2 The abstract summarizes and reflects the work described in the manuscript.

3 The key words reflect the focus of the manuscript.

4 The manuscript describes the background, and present status of the topic.

5 The manuscript gives an overview of the current literature in the field. It cites important and authoritative references.

6 The research objectives were reasonably achieved.

7 The manuscript is well, concisely and coherently organized and presented. It is well written, the style, language and grammar are good.

8 The authors prepared the manuscript according to appropriate reporting.

9 The manuscript meets the requirements of ethics. 

As COVID-19 might directly cause a high number of complications in patients with pre-existing chronic liver disease, increasing their risk of hepatic decompensation, and also has indirect consequences in the management of patients with liver disease, especially in those suffering from decompensated cirrhosis and HCC, as well as concerning the follow-up and the availability of therapeutic possibilities, the aim of this narrative review was to summarize the currently hypothesized pathophysiological mechanisms of liver injury in patients with COVID-19, mediated by both direct virus invasion and indirect effects, and to analyze the consequence of the pandemic in patients with chronic liver disease and liver tumors. A particular focus was aimed to the implemented management strategies.

The authors have written an interesting narrative review that summarizes the current state of knowledge with much speculation and little evidence, citing the literature available to date. Actually it is well written. More emphasis should be put on direct and indirect viral damage and drug-induced liver injury and the underlying mechanisms citing the literature available to date.

Author Response

Dear Reviewer,

Thank you for the opportunity to revise and improve our paper according to your comments.

We have modified the main text in accordance with your  insightful and significant suggestions and we have replied point by point to all requested revisions. 

We hope that now our manuscript reaches a suitable level for a possible publication on International Journal of Molecular Sciences

REVIEWER COMMENT:

1.The title is really striking and partly reflects the main subject of the manuscript.

2 The abstract summarizes and reflects the work described in the manuscript.

3 The key words reflect the focus of the manuscript.

4 The manuscript describes the background, and present status of the topic.

5 The manuscript gives an overview of the current literature in the field. It cites important and authoritative references.

6 The research objectives were reasonably achieved.

7 The manuscript is well, concisely and coherently organized and presented. It is well written, the style, language and grammar are good.

8 The authors prepared the manuscript according to appropriate reporting.

9 The manuscript meets the requirements of ethics. 

As COVID-19 might directly cause a high number of complications in patients with pre-existing chronic liver disease, increasing their risk of hepatic decompensation, and also has indirect consequences in the management of patients with liver disease, especially in those suffering from decompensated cirrhosis and HCC, as well as concerning the follow-up and the availability of therapeutic possibilities, the aim of this narrative review was to summarize the currently hypothesized pathophysiological mechanisms of liver injury in patients with COVID-19, mediated by both direct virus invasion and indirect effects, and to analyze the consequence of the pandemic in patients with chronic liver disease and liver tumors. A particular focus was aimed to the implemented management strategies.

The authors have written an interesting narrative review that summarizes the current state of knowledge with much speculation and little evidence, citing the literature available to date. Actually it is well written. More emphasis should be put on direct and indirect viral damage and drug-induced liver injury and the underlying mechanisms citing the literature available to date.

RE: Dear Reviewer 1, thank you for your appreciations and comments. As you have correctly underlined, the COVID-19 pandemic has profoundly strained health care resources around the world. Despite viral pneumonia represents the most common serious manifestation of COVID-19, the liver is now considered a secondary and often collateral target of this disease and its involvement can lead to important clinical consequences, both direct and indirect. However, the pathophysiological and immunological mechanisms of liver injury in patients with COVID-19 are still poorly understood, as well as their long-term sequelae, especially in those patients with an existing chronic liver disease. Therefore, clinicians must pay high attention to the most exposed and fragile patients and continuously search for new strategies that can be implemented in this new pandemic setting. As suggested by the Reviewer, to provide a clearer view of the pathophysiological mechanism of liver injury in patients with COVID-19, the section regarding the direct and indirect viral damage as well as the one regarding the drug-induced liver injury in these patients have been implemented. Moreover, the current literature available to date regarding the hypothesized underlying mechanisms of liver damage has been added. This contents have been also emphatyses in the Conclusion.

Sincerely,

Nicolò Brandi

Department of Radiology, IRCCS Azienda Ospedaliero-Universitaria di Bologna, Via Albertoni 15, Bologna, Italia.

Reviewer 2 Report

In this review authors illustrated the currently hypothesized pathophysiological mechanisms of liver injury in patients with COVID-19, mediated by both the direct invasion of the virus and its indirect effects. Moreover, authors analyzed the consequence of the pandemic in patients with chronic liver disease and liver tumors.

The manuscipt is interesting and generally well written. Only minor point could be improved. In particular:

Lines 36-38: it deserves to be mentioned that this virus can also lead to non restiratory diseases such as Preeclampsia,  male infertility and brain damage (as recently reviewed PMID: 35114008, 35943095, 32934351). This is an important point to add since it can give more importance to the pleiotropic effect of this virus.

it would be useful adding a summary table or figure resumming the interesting outcome of COVID-19 in liver damage 

Author Response

Dear Reviewer,

Thank you for the opportunity to revise and improve our paper according to your comments.

We have modified the main text in accordance with your  insightful and significant suggestions and we have replied point by point to all requested revisions. 

We hope that now our manuscript reaches a suitable level for a possible publication on International Journal of Molecular Sciences

REVIEWER COMMENT:

In this review authors illustrated the currently hypothesized pathophysiological mechanisms of liver injury in patients with COVID-19, mediated by both the direct invasion of the virus and its indirect effects. Moreover, authors analyzed the consequence of the pandemic in patients with chronic liver disease and liver tumors.

The manuscipt is interesting and generally well written. Only minor point could be improved. In particular: 

Lines 36-38: it deserves to be mentioned that this virus can also lead to non restiratory diseases such as Preeclampsia,  male infertility and brain damage (as recently reviewed PMID: 35114008, 35943095, 32934351). This is an important point to add since it can give more importance to the pleiotropic effect of this virus.

RE: Dear Reviewer 2, thank you for your appreciations and comments. Despite viral pneumonia representing the most common serious manifestation of this viral infection, extrapulmonary manifestations of COVID-19 have progressively gained attention due to their links to clinical outcomes and their potential long-term sequelae, especially in critically ill patients. As you rightly pointed out, COVID-19 could lead to cardiovascular complications, renal injury, gastrointestinal symptoms, neurologic complications, dermatologic conditions as well as male infertility and pre-eclampsia. The pleiotropic effect of this virus has now been emphasized in the text, also providing the suggested literature.

REVIEWER COMMENT:

it would be useful adding a summary table or figure resumming the interesting outcome of COVID-19 in liver damage 

RE: Dear Reviewer 2, thank you for your insightful suggestion. As you have required, a Table resuming the main effects of liver damage due to COVID-19 infection in patients with pre-existing liver disease has been provided (Table 1).

Sincerely,

Nicolò Brandi

Department of Radiology, IRCCS Azienda Ospedaliero-Universitaria di Bologna, Via Albertoni 15, Bologna, Italia.